# Collection of Anthropogenic Litter from the Shores of Lake Malawi: Characterization of Plastic Debris and the Implications of Public Involvement in the African Great Lakes

**DOI:** 10.3390/toxics7040064

**Published:** 2019-12-13

**Authors:** Bahati S. Mayoma, Innocent S. Mjumira, Aubrery Efudala, Kristian Syberg, Farhan R. Khan

**Affiliations:** 1Department of Biology, University of Dodoma, Dodoma PO Box 338, 255, Tanzania; 2Beach and Underwater Cleanup Malawi, Lumbadzi PO Box 174, 265, Malawi; beach.underwatercleanupmalawi@gmail.com (I.S.M.); efudalaaubrey@gmail.com (A.E.); 3Department of Science and Environment, Roskilde University, Universitetsvej 1, PO Box 260, DK-4000 Roskilde, Denmark; ksyberg@ruc.dk (K.S.); frkhan@ruc.dk (F.R.K.)

**Keywords:** plastics, plastic debris, African great lakes, freshwater, beach clean-up, citizen science

## Abstract

Anthropogenic debris is an environmental problem that affects beaches and coastlines worldwide. The abundance of beach debris is often documented with the use of public volunteers. To date, such community participations have been largely confined to the marine environment, but the presence and impact of anthropogenic debris on freshwater shorelines has been increasingly recognized. Our study presents the first such information from the African Great Lakes, specifically Lake Malawi. A total of 490,064 items of anthropogenic litter were collected by over 2000 volunteers in a clean-up campaign that took place annually between 2015 and 2018. Approximately 80% of the anthropogenic debris was comprised of plastic litter, with plastic carrier bags being the most common item. The dominance of plastic litter, and in particular the presence of plastic bags, which have subjected to bans in some African countries, is discussed. The broader implications of citizen science in the African Great Lakes area is also discussed.

## 1. Introduction

Anthropogenic debris is an environmental problem that affects beaches and coastlines worldwide [1,2]. Typically consisting of larger items (> 2 cm) of plastics, glass, metal, paper, and wood, plastic litter is often recorded as the most abundant type [3,4]. Although the presence of litter may decrease the tourist potential of affected beaches [5], anthropogenic debris is not a purely aesthetic issue, as it also has ecological consequences and is detrimental to wildlife. Debris, particularly plastics, pose a risk of entanglement to aquatic animals, including fish and birds, and are also likely to be mistakenly ingested owing to their similarity to food items [6,7]. Plastics can also act as vectors transporting adsorbed chemical (e.g., hydrophobic persistent organic pollutants (POPs) or trace metals) or biological (e.g., microorganisms) loads into new geographical regions [8].

The documenting of beach debris has taken place globally on marine shorelines. Examples include clean-ups and surveys on beaches along the Pacific coast of Chile [3], in Greece on the Mediterranean Sea [9], in Orange County in California (USA, [10]), and a ten year nationwide assessment of anthropogenic litter on beaches of the British Isles [11], to name a few. These studies all have in common the utilization of volunteers in their collection of data during the beach survey and clean-up activities. In fact, over the last 10 years, the number of volunteers taking part in clean- ups has doubled from about half a million people in 2009 to just over a million in 2019 [12,13]. The use of volunteers allows for extensive sampling at a wider range of sites [14] and although data validity may be open to question, there is no statistical difference between the data obtained from volunteer clean-ups and those performed by experienced surveyors, provided a clear protocol is followed [15]. Furthermore, community participation and citizen science are considered to be effective methods of increasing public awareness of environmental challenges and has been particularly well used with plastics [16,17].

To date, such community participations have been largely confined to the marine environment, but the presence and impact of anthropogenic debris on freshwater shorelines has been increasingly recognized, with a handful of surveys and clean-ups conducted along the shoreline of the North American Great Lakes [18]. However, such information remains scarce for freshwater environments and has not yet been available for the African Great Lakes. This study presents, for the first time, data on the anthropogenic debris collected by volunteers on the Malawian coast of Lake Malawi. Lake Malawi is bordered by Malawi, Tanzania, and Mozambique in the East African rift valley region (Figure 1). Referred to as Lake Nyasa in Tanzania and Lake Niassa in Mozambique, Lake Malawi is the third largest lake on the African continent and the world’s ninth largest lake overall (surface area: 29,500 km^2^, mean depth: 264 m, maximum depth: 700 m, 19). Together with Lakes Victoria and Tanganyika, Lake Malawi holds approximately 25% of the world’s total surface freshwater supply [19]. Lake Malawi is said to be the most species-rich lake in the world, with an estimated 500–1000 species of fish present in the lake [19]. Perhaps unsurprisingly, approximately 70% of the animal protein in the Malawian diet is in the form of fish [19]. The presence of debris, particularly plastics, on the shoreline of Lake Malawi could result in amounts of microplastics (MPs, < 5 mm) entering the water via degradation and being ingested by the fish. The ingestion of plastic particles by fish has been well documented globally from numerous aquatic habitats (e.g., [20,21,22,23]), including the African Great Lakes [24]. In this regard, understanding the land-based debris, as is the case with beach clean ups, may be an important step in understanding what enters the water.

In general, there is a scarcity of information regarding the presence and abundance of plastics (and microplastics) in Africa and specifically within the African Great Lakes [25], but two studies from Lake Victoria document the scope of the potential problem. Solid waste anthropogenic inputs were investigated in the Tanzanian waters of Lake Victoria by sampling across three main ecological zones; the nearshore (0–20 m depth), intermediate zone (20–40 m depth), and deep offshore waters (> 40 m) [26]. Plastic debris was found at all depths, with the dominant waste types originating from fishing activities; multifilament gillnets (44% of all debris), monofilament gillnets (42%), longlines and hooks (7%), and floats (1%). Plastic bags (4%) and clothing (2%) accounted for the remaining solid waste [26]. The second study was also conducted in the Tanzanian waters of Lake Victoria around the urban center of Mwanza and showed that two fish species, Nile perch (*Lates niloticus*) and Nile tilapia (*Oreochromis niloticus*), contained MPs within their digestive tracts [24]. In total, suspected plastic particles were recovered from the gastrointestinal tracts of 11 perch (55%) and 7 tilapia (35%), with spectral analysis confirming the presence of MPs in 20% of each species [24]. Together, these two studies provide evidence that plastic debris in Lake Victoria is subject to degradation and the products of that breakdown are available for ingestion by resident fish populations [25].

In the present, relatively small-scale, study, anthropogenic debris was surveyed and collected from three sites over a four-year period (2015–2018 inclusive) located in Central and Northern regions of Malawi. The clean-up sites represent the busiest beaches along Lake Malawi. In total, over 2000 volunteers took part in the clean-up activities, which were directed by Malawi beach and underwater clean-up, an environmental NGO. The protocol to instruct the volunteers was that of the ocean conservancy clean-up, which has been widely used in coastal locations. The data presented here adds to the little information on anthropogenic and plastic debris collected from freshwater shorelines and specifically is the first of its kind from the African Great Lakes. 

## 2. Materials and Methods 

### 2.1. Study Sites

The present study was conducted along the Malawian shorelines of Lake Malawi. Tourism, transportation, trade, agriculture, and fisheries related activities are found within a growing number of towns along the shoreline. Several district townships have hydrological linkage with the lake shoreline through rivers, stream, and storm waters. The most notable townships in Malawi within proximity of Lake Malawi (estimated population from 2018 census) are: Lilongwe (> 1,630,000), Mangochi (> 600,000), Salima (> 470,000), Karonga (> 365,000), and Nkhata bay (> 285,000). Three beaches were selected for clean-ups based on their proximity to urbanization, tourism, fishing, commerce, and transportation. Kambiri beach (34.617952°, –13.781318°) and Mpatsa beach (34.602600°, –13.762000°) are located towards the southern end of Lake Malawi, whereas Nkhata jetty (34.290731°, –11.606589°) is more northernly (Figure 1). Whilst the two beaches are favorite tourist destinations and also home to fishing and farming activities, Nkhata jetty forms a strategic port hub linking Malawi with the other countries that border Lake Malawi (Table 1). The port has various socio-economic activities, including construction, transport, and trade.

### 2.2. Beach Clean-Up Methodology

Beach clean-ups were conducted each September from 2015 to 2018 during the dry season. Clean-ups were organized by the ‘beach and underwater clean-up’, an NGO based in Lumbadzi, Malawi. The timing coincided with that of the ocean conservancy’s international coastal clean-up day, an annual global event which takes place in September and October. Each year only one beach was selected, as follows: Kambiri beach in 2015 and 2016, Mpatsa beach in 2017, and Nkhata jetty in 2018 (Table 2). 

Upon arrival, each volunteer was registered and provided with clean-up equipment such as gloves, collection bags, and gumboots. At each site, the name, location, time, distance, and number of volunteers were recorded in a pre-designed data collection form according to Lewis (2002) [27]. During clean-up, a stretch of beach measuring 100 m in length and 20 m in width was identified and cleaned before moving to the next stretch. Litter was collected from the shallow water to the highest water strandline. This was done for a maximum of four hours depending on weather conditions on the day and the number of volunteers. A total of 2139 volunteers took part in cleanup as follows: 350 and 821 people in Kambiri beach in 2015 and 2106, respectively, 64 people in Mpatsa beach in 2017, and 904 people in Nkhata jetty in 2018 (Table 2).

The methodology for clean-up events was adopted from the international coastal cleanup protocol [12,13]. The protocol constitutes 19 individual steps which were followed in three phases of the clean-up: (1) before clean-up, (2) during clean-up, and (3) immediately after clean-up. The steps included the identification of safe collection sites and coordinators, and training of volunteers before the clean-up; dealing with entangled or injured animals and filling in data cards during the clean-up; and recycling and disposal after clean-up. Additionally, the compilation of clean-up data was performed. 

Litter larger than 2 cm was collected by hand, enumerated, and then classified into categories: glass, metal, paper, construction material, and plastic [28]. For easy sorting of recyclable trash, volunteers were encouraged to work in teams of five people. Each volunteer was given one trash bag designated for the major groups of recyclable waste (e.g., plastic bottles, glass, aluminum) which were sorted as they went and recorded on the data card by the group leader. Other litter categories not suitable for recycling, such as plastic carrier bags, cigarette butts, fishing gears, and personal hygiene products, were identified, counted, and collected for disposal. After the clean-up, all recyclable items were transported to the recycling centers while the rest of the anthropogenic litter was transported to designated waste sites.

### 2.3. Data Handling

Litter items in each major category (glass, metal, paper, construction material, and plastic) were compiled and totaled, as described in the ocean conservancy protocols. Plastic litter was further divided as carrier bags, personal hygiene products, beverage bottles, fishing gear, cups, packaging materials, bottle caps, lids, food wrappers, cutlery, cigarette butts, tires, and straws/stirrers.

## 3. Results and Discussion 

### 3.1. Collection of Anthropogenic Litter

A total of 490,064 items of anthropogenic litter were collected by over 2000 volunteers at the three locations over the four-year period. With reference to the number of items collected, Nkhata jetty (>200,000 items) was the most impacted site, followed by Kambiri beach (2016 (> 175,000 items) and 2015 (approximately 90,000 items) and the Mpatsa beach (approximately 20,000 items) (Table 2). Typically, the numbers of items collected at different locations are normalized to the area of the site in order to make more valid comparisons of litter densities. The order of sites with regard to litter density is: Nkhata jetty (5.11 items/m^2^) > Kambiri beach (2016, 1.11 items/m^2^) > Mpatsa beach (0.96 items/m^2^) > Kambiri beach (2015, 0.89 items/m^2^). These densities are marginally lower than those reported elsewhere, 2 to 10 items/m^2^ along the coast of Chile [3] and 2.3 to 6.3 items/m^2^ reported along Brazilian coast [4], but typical densities reported globally range from 0–2–5 items/m^2^ [3]. An average density of 45,000 items/m^2^ was recorded from Japanese beaches [29], though it should be noted that the authors counted fragmented Styrofoam as individual items which are not usually counted individually in other studies.

The apparent greater levels of litter at Nkhata jetty beach may be attributed to its function as a busy port which links Malawi to Tanzania. Ports and associated activities have been reported to cause high level of debris in their surrounding environment [30]. Conversely, Kambiri beach and Mpatsa beach have relatively little industrial usage, instead being utilized for tourism, fishing, and farming. However, such results need to be interpreted with caution, as the number of items found at each site was significantly correlated to the number of volunteers conducting the clean-up. Correlating the number of volunteers to total litter found (values presented in Table 2) resulted in an R^2^ value of 0.995 (*p* = 0.00235, R program). Thus, it is difficult to make conclusive comments regarding the relative abundance and density of anthropogenic debris at the different sites on the Lake Malawi shoreline when citizen participation appears to be such an influential factor, but certainly the greater the number of volunteers, the more litter is likely to be collected. Furthermore, in this case where volunteer numbers greatly influenced the number of items collected, we refrain from further discussing densities, instead using the percentages of items in each category for comparative purposes.

The composition of debris across all beaches was plastic 80.2 ± 5.0%, metal 9.5 ± 2.5%, glass 4.9 ± 3.8%, paper 4.6 ± 4.0%, and construction material 0.9 ± 1.5%. Moreover, the percentages of each category remained relatively consistent across all sites, with plastic litter being by far the most abundant type of anthropogenic debris (Figure 2). Again, these results are in keeping with studies from other locations. European, North, and South American clean-ups have also reported plastics as the main constituent of anthropogenic litter on coastal beaches [3,9,10,11]. Whilst little data is available for freshwater shorelines, volunteer beach clean-ups along the North American Great Lakes revealed that typically more than 80% of anthropogenic litter is comprised of plastics [18]. Our study adds data from the African Great Lakes to the global pattern of plastics as being the dominant litter type along freshwater and marine shorelines.

### 3.2. Plastic Litter

As with anthropogenic debris, in general the composition of plastic debris was also relatively consistent across sites: plastic carrier bags (22.5–49.4%), personal hygiene products (7.15–18.6%), beverage bottles (1.9–18.8%), fishing gears (0.21–20.25%), cups (1.9–13.3%), packaging materials (1.3–13.8%), bottle caps (4.4–11.6%), lids (1.0–8.8%), and food wrappers (4.4–11.5%) (Figure 3). Remaining categories constituted less than 2% of plastic collected. The trend observed at Lake Malawi differs from that reported by the 2019 ocean conservancy annual report (13), which found that plastic bags are the sixth most common item found, whilst personal hygiene products (e.g., condoms, diapers, syringes), which rank second at Lake Malawi, are not within the top 10 items found on coastal beaches. In its annual clean-up report of 2010, the ocean conservancy [12] found that personal hygiene products contributed to only 1% of plastic items collected in African clean-ups. 

Similar findings of plastic litter have also been reported elsewhere. The shoreline of the remote freshwater lake, Lake Hovsgol in northwest Mongolia, was dominated by plastics bags, beverage bottles, and discarded fishing gear [31]. Moreover, plastic bags constituted a small percentage of debris trawled from Lake Victoria [26] and the microplastic polymers found in the gastrointestinal tracts of fish sampled from Lake Victoria may have originated from this source [24], suggesting that shoreline waste may end up as the microplastics in the adjacent lake [25]. Lessons from Lake Victoria could be transferrable to Lake Malawi given the abundance of discarded plastic bags.

The issue of plastic bags is particularly prescient for Africa, as several countries have proceeded to reduce or ban plastic bag use. Such measures have been made on the grounds of environmental and public health, as discarded plastic bags have been shown to block gutters and drains which create storm water problems and collect water, which provides a breeding ground for mosquitos that spread malaria, and the use of bags as toilets has been linked to the spread of disease [32,33]. The government of South Africa introduced levies on plastic bag use in 2003 [34], and in Rwanda in 2005, a ban on the use and importation of plastic bags of < 100 microns thick was imposed [33]. Tanzania having made a similar ban based on thickness in 2006 has now, in June 2019, implemented a complete ban on manufacturing and importing plastic bags into country [35]. Developments in Tanzania may impact the amount of plastic waste entering Lake Malawi and may then also influence the policies of Malawi.

### 3.3. Implications for Public Involvement and Plastic Pollution in the African Great Lakes

The present study was the first to document the results of beach clean-up activities in the freshwater African Great Lakes. Though limited in size and scope, a pattern emerged that confirms previous coastal findings that plastic litter is the dominant type of anthropogenic debris. This study highlights the role that volunteers can play in both the remediation of the environment and the collection of scientific data. Citizen science has become a widely used initiative in combating plastic pollution. Organizations such as the national oceanic and atmospheric administration (NOAA) in the United States have developed a mobile phone application called “Marine Debris Tracker app” together with Southeast Atlantic marine debris initiative (SEA-MDI), which allows the public to report findings of litter from beaches and waterways [36]. Similarly, in the EU, the European environmental agency (EEA) has developed the “Marine LitterWatch” (MLW) program which collects beach litter data from both the monitoring efforts of the authorities and citizen science projects [37]. The MLW program provides scientific input to the EU policy process and thus illustrates the regulatory potential of these programs. The adoption of similar technologies may also aid the mapping of litter in other areas, and particularly in the African Great Lakes region, where information is scarce.

An interesting finding of this study is the strong positive correlation between volunteer numbers and total debris collected. This may suggest that not all litter was collected at each site, which may be due to the fixed time restrictions placed on the clean-up (4 h). Under these conditions, greater public involvement would lead to better results, but it should be noted that the number of sites is too low to make any conclusive statements on this relationship. One grouping that has been shown to be particularly valuable are students. Students can play an active role in collecting and monitoring data using mobile applications, such as the one made by NOAA. In just one example from the Roskilde Fjord region in Denmark, students collaborated with scientists to produce data on the occurrence of marine litter at 12 beaches around the fjord [17]. The students analyzed the data using a protocol inspired by the marine litter watch protocol and were shown to be able to follow instructions and generate reliable data [17]. The involvement of students in collecting data serves as an example of transformative learning [38] and helps to raise public awareness in general, and particularly in the next generation. Another advantage of involving students and schools is that by educating teachers on how to sample properly, they can ensure that the scientific protocol is properly followed.

Even though the described cases are from Europe and the United States, there is equal potential in other places, such as within Africa. It is not correct to say that African nations and people are unaware of the issues surrounding plastic pollution. Currently, Africa has the highest percentage of countries (∼46%) with plastic bans [39] and numerous stakeholders are involving themselves in the fight against plastic pollution. Policy makers have enacted banning and reduction legislation and new technologies are being adopted across the plastics supply and recycling chain to enhance the possibilities of effective closed-loop waste management [40]. Clean-up activities, such as the one reported here from the shorelines of Lake Malawi, can be used to remove anthropogenic debris and raise public awareness.

## Figures and Tables

**Figure 1 toxics-07-00064-f001:**
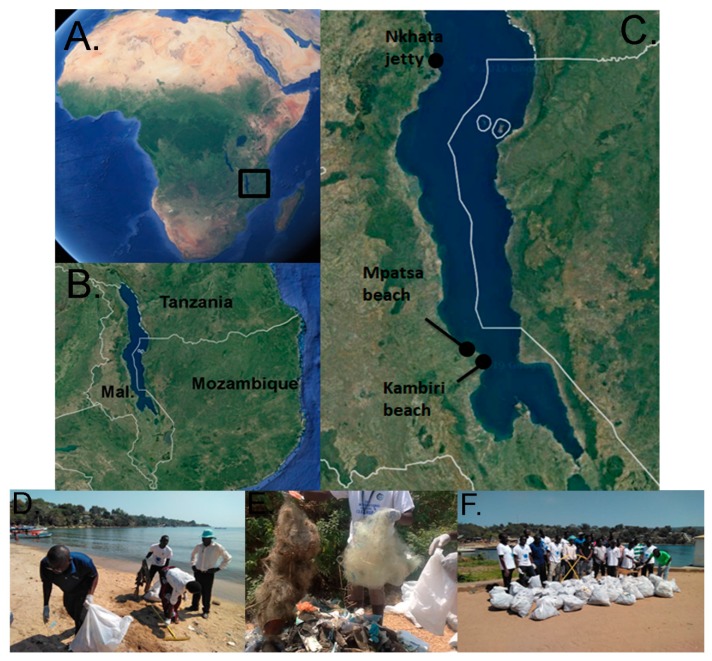
Location of the three beach clean-up sites on the Malawian coast of Lake Malawi (**A**–**C**, Mal. = Malawi). Images **D**–**F** show the volunteers during the clean-up and typical examples of the debris collected during the clean-up at Nkhata jetty in 2018.

**Figure 2 toxics-07-00064-f002:**
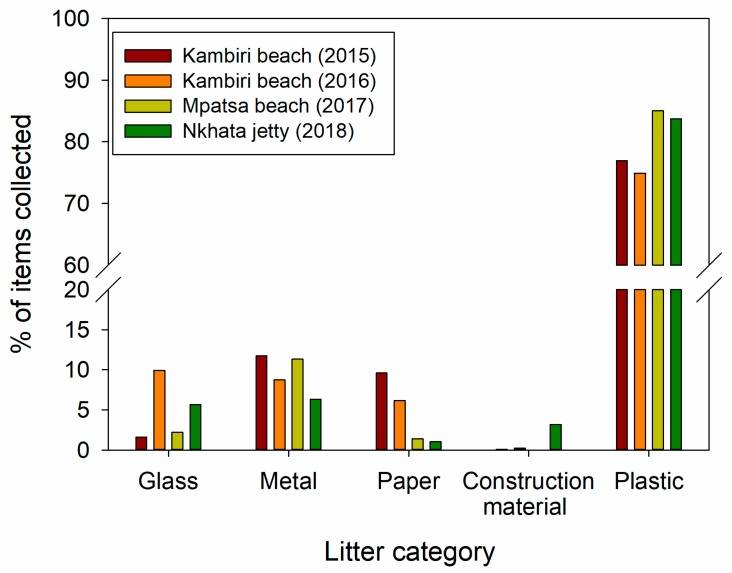
Percentage of items found of different categories of anthropogenic material (glass, metal, paper, construction material, and plastic) across sites and clean-up years.

**Figure 3 toxics-07-00064-f003:**
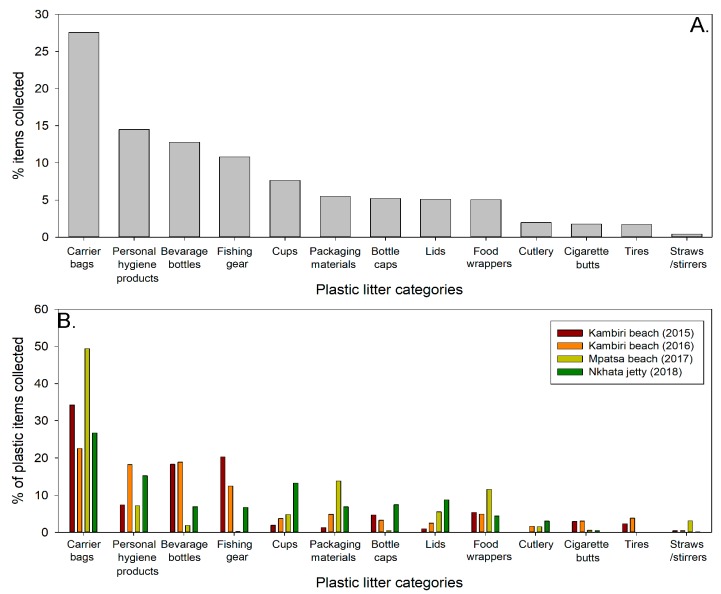
Breakdown of different categories of plastic debris as percentage of plastic items found at all sites and years combined (**A**) and by each site and year (**B**).

**Table 1 toxics-07-00064-t001:** Site descriptions.

Site	Location (Longitude, Latitude)	Description and Main Anthropogenic Activities
Kambiri beach	34.617952°, −13.781318°	Sandy beach with relatively gentle slope and scattered human settlements Tourism, maize farming, fishing, open market, livestock present, transportation mostly by wooden boats
Mpatsa beach	34.602600°, −13.762000°	Sandy beach with relatively gentle slope and scattered human settlements Tourism, fishing, maize farming, transportation mostly by wooden boats
Nkhata jetty	34.290731°, −11.606589°	Port area which forms a transportation hub linking Malawi, Tanzania, Mozambique and inland islands; historical memorial site which attracts tourists, recreation, cassava farming, fishing and market. Beach with minimal sand accumulation. Has various infrastructure developments including warehouse to cater for imports and exports services.

**Table 2 toxics-07-00064-t002:** Details of beach clean-up campaigns at each site and year.

Site	Years	Approximated Area Covered (m^2^)	No. of Volunteers	Total No. of Items Collected	Total No. of Plastic Items Collected
Kambiri beach	2015	100,000	350	89,442	68,836
	2016	160,000	821	177,087	132,666
Mpatsa beach	2017	20,000	64	19,238	16,361
Nkhata jetty	2018	40,000	904	204,297	171,092

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
