# Peer review of "Collection of Anthropogenic Litter from the Shores of Lake Malawi: Characterization of Plastic Debris and the Implications of Public Involvement in the African Great Lakes"

_toxics, 2019, doi:10.3390/toxics7040064_

Round 1

Reviewer 1 Report

Mayoma and colleagues describe the collection of debris from the shores of a freshwater body (Lake Malawi) and the subsequent characterization of the collected debris. The manuscript is well written and the methodology used is well referenced. I particularly enjoyed the attention that the authors placed towards the number of volunteers and the total volume of litter found/collected (Lines 174-183). This was a sober look at the gathered data, which the authors could have tried to extrapolate to other variables (such as population densities), but that the authors – in my opinion, correctly – did not. Hence, considering the aforementioned observations, minor modifications, mostly referring to format, are recommended before acceptance.

Introduction:

“the problem of anthropogenic debris is not purely aesthetic, although the presence of litter may decrease the tourist potential of affected beaches (5), as it is also has ecological consequences and is detrimental to wildlife” – This sentence is confusing. Please re-phrase.

“transporting adhered chemical” – “adsorbed” is perhaps more adequate.

Should read “Lates niloticus” and “Oreochromis niloticus”.

Bibliographical referencing is not homogeneous throughout the manuscript; sometimes, it is numbered; other times, it is in the author-date format. Please correct.

“In the present, relatively…” – In the present “study/work”?

“The shoreline is characterized by tourism, transportation, trade, agriculture, and fisheries related activities…” – Although it is understandable what the authors mean, the characteristics of the shoreline are not “tourism, trade”, etc. These are activities typically developed within that area.

Materials and Methods

“Each year only one beach was selected”. Why? The authors should explain the reasoning behind this, as the prevalence of debris could (it does) vary in time.

“Litters were collected from…” – Should read “litter was collected…”

“handpicked up” – Should read “collected by hand” or “manually collected” or some variation.

Results

“wastransported” – space is missing. This occurs in many sections of the text and should be corrected. Examples include: lines 163, 166, 199, 237, 248, 251… and in other sections.

Figure 2 – Please change “glasses”, to “glass”, “metals” to “metal”, “papers” to “paper” and “construction” to “construction materials”.

“As a country that shares the waters of Lake Malawi…” – This sentence requires some re-structuring.

Figure 3 – The classification of the items found should perhaps be clearer. For example, when referring to “tires”, do the authors mean portions of tires, or complete, discarded tires? Should also read “personal hygiene products” and “packaging materials”.

“Citizen science has become widely use…” – This sentence should be re-phrased.

“An interesting finding of this study is the strong positive…” – Yes, this is a rather interesting finding, but the authors should also mention that the number of “replicates” in which this was observed was also very small. This finding is based on four collection campaigns/activities. The identification (and validation) of such a correlation will require a comparison of a far greater number of activities. This, as noted, should be mentioned.  

“Arguably the largest group that needs exhibit change are the public and initiatives” – This sentence does not make a lot of sense. Please correct.

Reviewer 2 Report

This paper is interesting and the provided information can be useful for the researchers in this field. The paper needs some minor revisions as follows:

The language needs to be checked in whole the manuscript to remove grammatical errors. I suggest writing in the past tense not present. Make a separate separation for the conclusion. now is not clear. The addition of more analysis such as spatial analysis using Moran's I clustering in ArcMap or Generalised Linear Mixed Model to examine temporal patterns in the abundance of total litter.  Reference should be updated to more recent studies.
